# Steatotic Liver Disease in Older Adults: Clinical Implications and Unmet Needs

**DOI:** 10.3390/nu17132189

**Published:** 2025-06-30

**Authors:** Daniel Clayton-Chubb, William W. Kemp, Ammar Majeed, Peter W. Lange, Jessica A. Fitzpatrick, Karl Vaz, John S. Lubel, Alexander D. Hodge, Joanne Ryan, John J. McNeil, Alice J. Owen, Robyn L. Woods, Stuart K. Roberts

**Affiliations:** 1Department of Gastroenterology, Alfred Health, Melbourne 3004, Australia; w.kemp@alfred.org.au (W.W.K.); a.majeed@alfred.org.au (A.M.); jessica.fitzpatrick1@monash.edu (J.A.F.); karl.vaz@austin.org.au (K.V.); johnlubel@icloud.com (J.S.L.); s.roberts@alfred.org.au (S.K.R.); 2School of Translational Medicine, Monash University, Melbourne 3004, Australia; 3Department of Gastroenterology, Eastern Health, Box Hill 3128, Australia; alex.hodge@monash.edu; 4Department of Geriatrics and General Medicine, Werribee Mercy Hospital, Werribee 3030, Australia; aprofpeterlange@outlook.com; 5Department of Aged Care and Medicine, The Royal Melbourne Hospital, Melbourne Medical School, The University of Melbourne, Parkville 3050, Australia; 6Department of Gastroenterology, Northern Health, Epping 3076, Australia; 7School of Health and Biomedical Science, RMIT University, Melbourne 3083, Australia; 8Eastern Clinical School, Monash University, Box Hill 3128, Australia; 9School of Public Health and Preventive Medicine, Monash University, Melbourne 3004, Australia; joanne.ryan@monash.edu (J.R.); john.mcneil@monash.edu (J.J.M.); alice.owen@monash.edu (A.J.O.); robyn.woods@monash.edu (R.L.W.)

**Keywords:** ageing, older adults, liver, NAFLD, MASLD, metabolic dysfunction-associated steatotic liver disease, hepatology, cardiovascular disease, lifestyle

## Abstract

Metabolic dysfunction-associated steatotic liver disease (MASLD) is the commonest cause of chronic liver disease worldwide. Its incidence has been increasing rapidly, alongside the growing epidemics of type 2 diabetes mellitus and overweight/obesity. Global population age has also been increasing in parallel, and predictions indicate there will be more than 2 billion persons aged over 65 by the year 2050. The interplay between MASLD and other health conditions of older persons has been a focus of recent research. In this narrative review, we aim to describe its prevalence; clinical and sociodemographic associations; and outcomes for older persons, all of which are of significant importance when considering public health messaging as well as screening and counselling individual older adults.

## 1. Introduction

### 1.1. Background and Aetiology

Hepatic steatosis—defined as intrahepatic triacylglycerol rich lipid droplets of at least 5% of liver weight or 5% of hepatocytes containing lipid vacuoles—was first described by Thomas Addison in 1836 [1] in the context of excessive alcohol intake. It took another 150 years for non-alcoholic fatty liver disease (NAFLD) [2] and non-alcoholic steatohepatitis (NASH) [3] to be recognised as distinct disease entities unrelated to alcohol intake, and instead associated with insulin resistance, obesity and type 2 diabetes mellitus (T2DM) [4]. Since this time, there has been increased recognition of NAFLD, rising in parallel with the global epidemics of obesity [5] and T2DM [6], such that it is estimated that up to 38% of people globally are living with NAFLD [7].

To better recognise the link between non-alcoholic hepatic steatosis and aspects of the metabolic syndrome [8,9], a global Delphi process recently reclassified NAFLD to metabolic dysfunction-associated steatotic liver disease (MASLD) [10]. This new definition utilises similar criteria to the previous NAFLD definition—hepatic steatosis in the absence of excessive alcohol intake or steatogenic medication use—with the addition of at least one positive metabolic criterion (including dyslipidaemia, overweight/obesity, impaired fasting glucose/T2DM, or hypertension) [10]. The change in nomenclature has had minimal impact on disease epidemiology, with ~99% of NAFLD cases re-classified as MASLD [11,12]. The metabolic diseases used as criteria are not only important for linking disease aetiology with nomenclature—they also impact risk stratification and outcomes. It is well recognised that those living with both MASLD and T2DM have worse outcomes than those with either MASLD or T2DM alone [13,14]. MASLD is causally linked with a multitude of additional metabolic diseases, including hypertension [15] as well as dysregulated lipid metabolism [16].

Regardless of definition, MASLD represents a significant global burden of disease. Along with its more rapidly progressive form, metabolic dysfunction-associated steatohepatitis (MASH), MASLD is the most common chronic liver disease worldwide [17], and consequently an increasingly important cause of liver-related morbidity and mortality [18,19,20]. In Australia, it is estimated that from 2019 to 2030 there will be an 85% increase in MASLD liver deaths [21]; similar rates are predicted in the USA [22], with a tripling of MASLD-related deaths from 2020 to 2050. While these outcomes are linked to liver fibrosis resultant from the progression of liver diseases [23,24,25,26], the number of MASLD cases is increasing so rapidly that even low rates of progressive fibrosis will be associated with significantly increasing negative health outcomes worldwide. In spite of this, a recent study evaluating global preparedness for this rapidly increasing disease showed that none of the 102 countries evaluated had a national or sub-national strategy for MASLD, and almost one third of countries scored 0/100 on a preparedness index [27]. This raises significant concerns about the public health approach to this epidemic of chronic liver disease.

### 1.2. An Ageing Population

Rising in tandem with this epidemic of metabolic diseases is the ageing global population. It is estimated that, by 2050, one in six persons will be aged over 65 (compared to one in 11 in 2019), representing a doubling of older persons from 703 million to 1.5 billion globally [28]. This increase is driven by a combination of both reductions in premature mortality associated with non-communicable diseases (such as tobacco-related diseases and cardiovascular disease [CVD] [29]) as well as through improved sanitation [30,31] and increasing access to healthcare [32].

Despite this increasing number and proportion of older persons, societal ageism remains a significant barrier to older person care and community engagement [33]—and may contribute to the known relative lack of clinical trial research including older persons [34,35]. This is critically important when interpreting research in the context of population ageing, as without targeted studies, there is an inability to provide appropriate care and advice to this growing portion of the population. Importantly, without their participation in research, we also miss capturing what health outcomes are seen as important by the ageing population.

### 1.3. The Ageing Liver

The effects of ageing are not solely due to the accumulation of disease and the cumulative effect of risk factors; indeed, chronological age is a poor guide to ageing, and the ageing process is heterogenous between individuals and within the individual by organ system [36]. This accentuates the need for specific ageing-related research on common conditions such as MASLD, as generalizability from studies of younger adults is fraught.

Specific to the liver, there are anatomical and physiological alterations that result in functional changes and possible susceptibility to disease [37]. Hepatic blood flow and liver mass decreases, more so in females [38], and ageing is also associated with alterations in liver enzyme activity [39,40]. Senescent cells—cells that are non-functional but also not prone to apoptosis or necrosis—accumulate in a non-linear manner in the aged liver [41]. While these cells are non-functional, they are resistant to fibrosis [41]. In keeping with this understanding of hepatocyte senescence, prior work has shown that the ageing liver may also be characterised by fewer remnant functional hepatocytes, not just smaller liver volumes [42].

Adding further to this, it is known that ageing, while heterogeneously experienced, is not isolated to single organ systems; the bidirectional relationships between ageing organ systems leads to additional physiologic and functional changes [36].

Despite these known hepatic changes, there is still a relative paucity of liver-related research in older persons. While there have been some steps to address ageing-specific changes in other organ systems—for example, the development of a cardiovascular risk calculator specifically for people aged ≥65 years (SCORE2-OP [43])—further work is needed to improve our understanding of the whether there should be differences in the hepatological care we provide older persons. As such, this review aims to highlight the evidence—and gaps—that exist regarding MASLD in older persons, as well as identify key unmet needs and opportunities for future research.

## 2. Materials and Methods

In order to develop an overview of the impact of MASLD on older persons and related outcomes, databases were accessed using multiple keywords in combination, including: geriatric, ageing, aged, and elderly; MASLD and metabolic dysfunction-associated steatotic liver disease; NAFLD and non-alcoholic fatty liver disease; MAFLD and metabolic dysfunction-associated steatotic liver disease; MASH and metabolic dysfunction-associated steatohepatitis; NASH and non-alcoholic steatohepatitis; non-invasive testing; lifestyle; diet and diet score; sociodemography; non-communicable diseases; exercise; death and mortality; cardiovascular disease; malignancy; frailty; physical disability; cognition, cognitive decline, and dementia; senescence. Studies were restricted to humans unless a non-human animal or other pre-clinical model was felt necessary to expand on concepts. Additional material was identified through the reference section of key papers and personal knowledge of the literature. Databases accessed included PubMed and Google Scholar. No date restrictions were imposed, and papers up to 22 June 2025 were included.

## 3. MASLD and Fibrosis Identification

MASLD is, at its core, a histologic diagnosis [44]. However, due to the inherent risks of obtaining liver tissue via invasive biopsy as well as sampling error and inter-/intra-rater reporting variability [45,46], liver biopsy is not commonly performed [17]. In older persons this is especially important, as they may have more risks associated with biopsy (such as increasing rates of anticoagulation use [47]), and are less likely to tolerate complications due to reduced physiologic reserve [48,49]. As such, surrogate measures exist, which can be broadly stratified into radiological and score-based options.

Radiologically, ultrasonography (USS) [50,51], computed tomography (CT) [50,52], magnetic resonance imaging (MRI) [50,52,53], and the USS-derived controlled attenuation parameter (CAP) measurement [54] can all be used to identify hepatosteatosis. While their accuracy compared to biopsy is imperfect, they are commonly used in clinical practice [17], and there are no age-specific reasons for reduced accuracy compared to younger persons. Additionally, a variety of new and device-specific radiological algorithms to improve USS accuracy are being developed [55], although their relative accuracy and applicability to older persons specifically is unknown.

Score-based options use a composite of laboratory tests, anthropometric variables, and clinical parameters, with differences in their ability to identify hepatosteatosis in older persons. Several scores exist and have been predominantly validated in middle-aged populations, including the Fatty Liver Index (FLI) [56], Framingham Steatosis Index (FSI) [57], Dallas Steatosis Index (DSI) [58], Hepatic Steatosis Index (HSI) [59], Lipid Accumulation Product (LAP) [44], Visceral Adiposity Index (VAI) [60] and the ZJU Index (ZJU) [61]. Of all these scores, FLI is the only one validated against ultrasound in older persons with NAFLD [62]. None have been validated against biopsy in older persons. The components of these scores can be seen in Table 1.

Our previous observational study, using the FLI as the ‘gold standard’ epidemiological score for older persons, given its previous validation [62] and recommended use in local guidelines [64], established FSI as the next most accurate score for identifying MASLD [63]. Other non-FLI scores have significant limitations in older persons, largely due to (a) their derivation populations (recognising that a healthy BMI range in older persons is different to that in younger persons [65] and is ethnicity-specific [66,67]), and (b) the recognised change in alanine aminotransferase (ALT) values, along with a changing ALT/aspartate aminotransferase (AST) ratio with increasing age [40,68].

Although liver biopsy remains the gold standard for identifying liver fibrosis (a major determinant of liver- and cardiovascular-related outcomes in MASLD and other liver diseases) [23,24,25,26], the emergence of non-invasive tools to assess fibrosis has transformed clinical practice. Such tests include radiology-based tests (magnetic resonance elastography [MRE] [69,70] and vibration controlled transient elastography [VCTE] [70,71,72,73]), simple laboratory-based scores (including the Fibrosis-4 [FIB-4]; BMI, AST/ALT Ratio, Diabetes score [BARD]; AST to Platelet Ratio Index [APRI]; NAFLD Fibrosis; BMI and AST [BIMAST]; Metabolic Dysfunction-Associated Fibrosis 5 [MAF-5]; and Steatosis-Associated Fibrosis Estimator [SAFE] scores) [74,75,76,77,78], composite scores such as the MEFIB (combining MRE with FIB-4) [79], and proprietary direct-fibrosis laboratory tests (such as the Enhanced Liver Fibrosis [ELF] test [80]). The components of these scores can be seen in Table 2.

However, much like the steatosis indices, these scores have almost exclusively been validated in middle-aged populations, or with a predominantly middle-aged cohort [77], without considering potential differences that could occur with age. Indeed, the FIB-4 was initially age-limited (validated only between 35–65 years), and a new threshold was subsequently proposed for use in older adults [81]. Despite this, studies have still shown poorer discriminatory ability in those aged over 65 [82]. Similarly, the SAFE score has been shown to be markedly less useful as a screening test in those aged 60–80 years compared with middle-aged adults [83], with similar results seen for several other composite scores [84]. Few studies have focused on validating the ELF score in older persons; one reported no significant difference in biopsy-stratified ELF scores in those older than 60 compared with those younger than 60 [85], although there was a non-significant increase in ELF values with increasing age within each fibrosis-stage stratum. In keeping with this, in the general population, there is an age-dependent increase in ELF scores [86,87]. TIMP-1 may change with differing extra-hepatic clinical and nutritional state [88], and there is a small trend to changing PIIINP levels with increasing age (though insufficient to alter reference intervals) [89]. Additionally, most non-proprietary scores use the ALT (which falls with age) [40,68], and three of the four also use platelets (which may change with age in a sex-dependent manner) [90]. Given this, it could be hypothesized these indices perform less well in older persons than in their middle-aged derivation cohorts, although, apart from an altered FIB-4 upper limit of normal, there is no reference to this in current guidelines [17,91].
**Recommendations for Future Research**Identify a preferred steatosis index that is older-person specific, rather than a re-purposed index based on middle-aged personsIdentify a preferred non-proprietary fibrosis index that is older-person specific, and consider re-evaluation of (a) the age variable and (b) ALT/AST cut-offs or ratios in pre-existing scoresSpecifically evaluate the utility of the ELF and similar proprietary scores in older personsEvaluate the utility of novel device-specific sonographic techniques for MASLD identification in older persons

## 4. Lifestyle and Sociodemographic Considerations

### 4.1. Diet and Lifestyle

MASLD is, at its core, a non-communicable disease (NCD) [92], intrinsically linked to at least three other major NCDs (CVD [93], cancer [94], and T2DM [95,96]). Like other NCDs, there are multiple lifestyle and environmental risk-factors associated with the development of MASLD, including unhealthy diets [97,98,99,100], infrequent physical activity and/or increased sedentary time [101,102,103], metabolic dysregulation-related conditions (including dyslipidaemia, insulin resistance and hypertension) [8,104,105], obesity [106,107], and even air pollution [108]. These considerations are even more important in the context of MASLD in older adults, due to the increasing prevalence of metabolic diseases in older persons [109], reduced physical activity compared to younger persons [110], and an increasing risk of poorer diet quality over time [111].

Many of these changes are due to a constellation of ageing-related processes, including dietary changes in the context of altered living circumstances, bereavement, isolation, and access to transportation [112], and reduced physical function due to co-morbid medical conditions, a lack of interest, a poor understanding of physical activity requirements, and musculoskeletal pain [113,114,115]. Guidelines to reduce the progression or prevent the development of MASLD in older persons need to focus on practical strategies to overcome these unique barriers to NCD reduction in older persons, with a focus on attainable dietary change, physical activity engagement, and appropriate management of co-morbid metabolic diseases.
**Recommendations for Future Research**Evaluate the role of healthy lifestyle promotion (diet, physical activity, social engagement) in ameliorating the development or progression of MASLD along with other co-morbid cardiometabolic conditions in older personsEvaluate both individual and structural barriers to the implementation of public policy-level interventions for non-communicable diseases, including MASLD, in older persons


### 4.2. Sociodemographic Influences

Many NCDs are considered to be related to both the aforementioned ‘lifestyle’ conditions and other social determinants of health, including socioeconomic disadvantage, psychosocial stress, educational attainment, and housing conditions [116,117], as well as ethnicity [118], indigenous status [119], and living remotely [120,121]. Many of these have been shown to be related to the presence of MASLD [122,123], including some in older persons specifically [124]. Despite this, a recent scoping review did not find any prospective studies evaluating social determinants of health in MASLD patients in the clinic [125]. Neither of the recent European [17] or North American [91] guidelines reference social determinants of health, the recognition of which has important public policy implications in the management of MASLD more generally—and in older adults specifically.
**Recommendations for Future Research**Prospectively evaluate the presence of and outcomes related to social determinants of health in older persons with MASLD to guide future targeted interventions


## 5. MASLD and Clinical Outcomes in Older Persons

### 5.1. MASLD and Cardiovascular Disease

Traditionally, CVD has been described as the most common cause of death amongst adults with MASLD [126,127]. However, a more granular evaluation of the data raises several questions about these conclusions. Firstly, in a recent meta-analysis, it was noted that many studies adjust for covariates that include dyslipidaemia (rather than individual cholesterol parameters) and the presence/absence of hypertension (rather than blood pressure in mmHg) [126]. However, this is not how usual cardiovascular risk assessment is carried out. Most risk calculators (including the older-person specific SCORE2-OP) [43] utilise total cholesterol, high density lipoprotein (HDL) cholesterol, and systolic blood pressure. In our study looking at rates of fatal and non-fatal major adverse cardiovascular events (MACE) in older adults, we found an ‘independent’ relationship between MASLD and MACE when adjusting for categorical risk factors—a relationship that was lost when including multiple known predictors of CVD in older persons [128,129]. These findings are concordant with another meta-analysis, whereby MASLD was not associated with CVD-related mortality [130]. Similarly, some of the more commonly cited MASLD-related single nucleotide polymorphisms (for example *PNPLA3*) seem to confer a reduced risk of CVD [131], implying heterogeneity in CVD risk for different patients with MASLD.

Given these results, it is recommended that older individuals with known MASLD undergo CVD risk-stratification using an age-appropriate CVD risk calculator. However, it is not clear that MASLD itself is an independent or synergistic risk for MACE in older persons, nor that targeting MASLD specifically would reduce atherosclerotic cardiovascular events or CVD-related deaths beyond concomitant improvements in other cardiometabolic risk factors.

In contrast, there are data to support an independent relationship between MASLD and incident atrial fibrillation (AF), including in our own ASPREE cohort [128]. While the Rotterdam study showed that liver stiffness (rather than steatosis alone) was associated with AF [132], this is contradicted by another study conducted in older persons [128] as well as by a meta-analysis of observational studies of more general cohorts [133]. While this does not imply a need for AF-related screening in older persons with MASLD, it is a risk that could be discussed in clinic, and self-reports of arrhythmia or tachycardia should prompt further investigation.
**Recommendations for Future Research**Assess the risks of MASLD on CVD outcomes in older persons including the impact of hepatic steatosis and fibrosis severity while adjusting for all known contributors to riskAscertain whether MASLD patients (irrespective of CVD risk profile) benefit from primary prevention for CVD with statins and other therapiesDetermine whether there are specific markers of excess CVD risk in older persons with MASLD (such as advanced fibrosis, or MASLD with specific monogenic/polygenic findings)


### 5.2. MASLD and Extra-Hepatic Malignancy

Extra-hepatic malignancy is a significant concern in persons of all ages with MASLD. While there are conflicting data around its role in overall cancer-related mortality in the general population [130,134], MASLD is thought to confer an increased risk of extra-hepatic malignancy in middle-aged persons, particularly cancers arising from the gastrointestinal tract, breast, and genitourinary systems [94]. However, there is less cancer-focused research in the older adult MASLD population. Of the 10 studies cited in a recent meta-analysis [94], none had an average participant age of ≥65 years. While one observational Japanese study focused on persons aged ≥60 years, it compared MASLD-related chronic liver disease vs. hepatitis C-related chronic liver disease, rather than older adults with MASLD vs. no-MASLD [135].

This remains an important knowledge gap in the literature. Screening programs for common cancers (such as colorectal cancer) often recommended an ‘end date’ based on age or are otherwise recommended to be individualised based on perceived risks and benefits [136,137]. While MASLD may confer an increased risk of colorectal cancer—and thus potentially imply increased benefit to screening in older adults with MASLD—it is also associated with an increased prevalence of frailty and future physical disability [124,138,139]. A better understanding of risks and benefits of screening for such malignancy in older persons specifically would help guide clinical decision making, inform guidelines, and educate patients.
**Recommendations for Future Research**Evaluate whether MASLD confers an increased risk of extra-hepatic malignancy in older personsEvaluate whether screening for extra-hepatic malignancy in older persons with MASLD yields meaningful improvements in quality or length of life and disability-adjusted life-years saved


### 5.3. MASLD and Diverticulosis

Colonic diverticulosis is extremely common—over two thirds of adults eventually develop it—and is associated with both increasing age and a ‘Western’ lifestyle [140]. Indeed, in the USA, one study has reported a prevalence of over 70% in persons aged 80 years or older [141]. As we have noted previously, MASLD is also a lifestyle-related disease, and there are reasons to believe there may be shared underlying aetiological factors and pathophysiological drivers [142,143]. Indeed, obesity (which is inextricably linked with MASLD [106,107]) is also linked with the development of diverticular disease [142]. Additionally, central adiposity and increased visceral adipose tissue are more strongly linked with diverticulosis than generalised overweight [142,144], and these differences are also seen in those with MASLD [145].

Another such shared pathway includes the gut microbiome. While a detailed exploration of the microbiome is outside the scope of this review, there are known links between gastrointestinal ‘dysbiosis’ and the development of metabolic disorders such as MASLD [146,147,148]. Similarly, there are alterations in the microbiome of people living with diverticular disease, and alterations also exist between those with and without complications associated with their diverticulosis (i.e., incidentally identified diverticulosis compared with symptomatic uncomplications or complicated diverticular disease) [142].

However, despite these links, it remains uncertain whether targeting MASLD in older persons reduces the risk of either developing diverticular disease or conversion from asymptomatic to symptomatic disease.
**Recommendations for Future Research**Evaluate whether treating MASLD in older persons reduces the risk of developing complicated diverticular disease


### 5.4. MASLD and Infectious Complications

It has been estimated that approximately one-third of deaths in older age are attributable to infections [149]. There are known changes to the immune system with ageing [150], including impaired memory B cells and antibody production [151], impaired T cell function (which may also predispose to autoimmune disease and cancer) [152], and impaired innate immunity [149]; all of these may increase an affected individual’s susceptibility to infection. Additionally, MASLD has been identified as being associated with an increased risk of serious infections [153], including in those with ‘only’ simple steatosis through to those with cirrhosis [154]. One of the larger studies evaluating this link found that, while the risk of infection did increase with age, biopsy-proven MASLD remained associated with an overall increased risk of infection [154]. While the mechanisms leading to this remain speculative, it is possible that a combination of chronic low-grade inflammation, a disrupted intestinal barrier function, and altered adaptive and innate immunity may contribute [155,156,157,158,159]. Indeed, immunosenescence has previously been linked to the progression of MASLD in those with T2DM [160], and more general cellular senescence is also linked with metabolic disorders [161]. Better understanding the potential dual impact of ageing and MASLD on immune function and predisposition to specific infections would assist with the clinical management of impacted individuals.
**Recommendations for Future Research**Determine the overall impact of serious infections on older adults with MASLDBetter understand the dual impacts of ageing and MASLD on immune function


### 5.5. MASLD Associations with Frailty and Physical Disability

Sarcopenia and frailty are common problems in advanced chronic liver disease across populations of all ages and are associated with increased risk of complications and short-term mortality [162,163,164]. This risk is also bidirectional—those with increased baseline frailty are at increased risk of incident MASLD, cirrhosis, and other liver-related events [165]. The relationship between MASLD and frailty appears to be independent of cirrhosis status, though is more severe in those with advanced chronic liver disease than those without [166].

The relationship between frailty and MASLD has been studied predominantly in older persons, with recent work building on that evaluating the impact of MASLD on longer-term physical function [138]—although the MASLD–frailty relationship was also seen in at least one middle-aged cohort [167]. Our paper showed a significantly increased risk of developing persistent physical disability (defined as the loss of one of the basic activities of daily living [ADL] for at least six months, or placement in a residential care facility due to the loss of an ADL), despite adjustment for multiple potential contributors, including gait speed and grip strength [138]. The mechanism underlying this is uncertain, although it may provide the opportunity for intervention if it were better understood. This may then allow for improving the health span and well-being of older persons with MASLD.
**Recommendations for Future Research**Determine whether MASLD is an epiphenomenon or causative factor in the development of frailty and persistent physical disability (e.g., through metabolomic factors)Determine if a targeted intervention for frailty improves quality of life or health span in older persons with MASLD


### 5.6. MASLD and Cognitive Outcomes

Multiple studies have examined dementia risk and brain-ageing in middle-aged persons with MASLD, often finding an increased risk of cognitive decline [168,169,170]. This relationship is less clear, however, in older adults. While one Swedish study (in which participants had a median age of 70 years) found a relationship between MASLD and dementia [171], this study did not adjust for baseline cognitive performance. Other studies involving older persons have suggested either no relationship between MASLD and dementia [172], or even an inverse relationship, whereby baseline MASLD was associated with a reduced risk of incident dementia [138,173].

The reason for this discrepancy is unclear; it is possible that ‘survivors’ to older age with MASLD have a less cognitively harmful phenotype of their liver disease. It is also possible that—given the known accelerated weight loss that occurs prior to and during the development of dementia [174,175]—even if MASLD increases the overall risk of dementia in older persons, their steatotic liver disease may have regressed prior to the formal diagnosis of dementia. In either case, when counselling older persons, although baseline cognitive function testing may assist with overall risk stratification, MASLD in isolation does not seem to clearly increase the risk of dementia.

Understanding this discrepancy in findings is important. It is well known that dementia is one of the greatest concerns people have as they age [176,177] and this fear can act as a barrier to being evaluated for cognitive decline, despite concerns patients or caregivers have [178]. While debate persists around the potential benefits and harms of screening for dementia [179,180], one strategy may be to specifically target at-risk groups. While various models exist for dementia screening [181], it remains to be seen whether there are specific subgroups of patients with MASLD who would potentially benefit from screening—and if screening the middle-aged, whether stopping as people transition to later life is appropriate.
**Recommendations for Future Research**Determine if there are differences between middle-aged and older MASLD that influence the development of cognitive decline/dementiaEvaluate (a) the cognitive decline in the middle-aged/older adult transition and (b) biomarkers of dementia progression in older persons with MASLD and no MASLD to assist in understanding causality/reverse causality in these groups


### 5.7. MASLD and Mortality

MASLD is frequently cited as a cause of increased mortality in middle-aged adults [134,182,183]. Given this, and other known complications of the disease, case-finding and consideration of pharmacologic intervention has been proposed in higher-risk groups [17]. However, there is less certainty that MASLD leads to increased mortality in older persons. Key studies have shown either no excess mortality risk [128,184] or a markedly attenuated risk [185] in older persons (depending on age cut-off). While the study based on the ASPREE (ASPirin in Reducing Events in the Elderly) cohort [128] may have some inherent bias due to exclusion of individuals with known pre-existing CVD and an *a priori* presumed low 5-year mortality risk (a key inclusion criterion for the study), this is precisely the group likely to be considered for MASLD screening. In contrast, the Rotterdam study did not have pre-specified exclusion criteria [186], and given the similar findings of all of these major studies, it is highly likely that MASLD does not lead to an excess of premature mortality in older adults. This has been further evaluated using genetic markers of MASLD rather than composite scores or radiology, with similar findings [187].

## 6. MASLD Treatments in Older Persons

### 6.1. Treatment Overview

A wide range of treatments are described in the literature and are well summarised in the recent North American [91] and European guidelines [17] as well as a recent review article focusing on liver-directed therapies [188]. While historically MASLD treatment has focused on (a) lifestyle interventions as well as (b) the management of known co-morbidities, there is now a therapy specifically licensed for the treatment of MASH—resmetirom [189]—with several more in the pipeline [188]. However, there are specific nuances to consider regarding both lifestyle and adjunctive pharmacotherapy, as well as the prescription of MASLD-specific medications in older persons, due to differences in ageing physiology, the risks of polypharmacy, and change in competing risks over time as the population ages. As such, here we describe key considerations around guidelines and recommendations that tend to be based on data from middle-aged persons and highlight opportunities for future research.

### 6.2. Lifestyle Therapies: Weight Loss and Diet

Weight loss features prominently in North American and European MASLD guidelines, with a target of 3–5+% to reduce steatosis and 7–10+% to improve MASH and/or fibrosis stage [17,91], particularly in those who are overweight/obese. These recommendations are primarily supported by studies in middle-aged adults. Multiple iterations of calorie-restricted diets have been studied in MASLD, with some benefits identified in a recent systemic review and meta-analysis (although none had a mean participant age over 55 years) [190]. Whilst these weight loss targets improve not only steatosis itself and also concomitant metabolic dysfunction (such as insulin resistance), they are challenging to achieve and maintain, with weight regain occurring slowly in both MASLD-focused studies [191,192] and in general weight loss studies [193]. Additionally, not all patients benefit from significant weight loss. In particular, in so-called ‘lean’ MASLD subjects, the target weight loss is capped at 3–5% [17], while in those with a background history of disordered eating patterns, implementation of weight loss measures requires a careful and balanced evaluation of the risks and benefits as well as additional psychological support [194]. Importantly, for the older population, there are also data that suggested they may be at increased risk of mortality as a result of significant weight loss [195], although the implications for this in the MASLD-group specifically are unknown.

An additional strategy compared to weight loss alone is to optimise overall diet quality, as well as macronutrient and micronutrient composition, for those with MASLD. One option is to institute a Mediterranean Diet pattern of eating, with a focus on high intakes of polyphenol-rich olive oil, vegetables, fruits, nuts/seeds, legumes, and seafood. It also focuses on a reduction in added sugar, refined carbohydrates, saturated fat, red and processed meats, and ultra-processed foods [196,197]. The Mediterranean Diet was shown in a 2021 meta-analysis to be associated with reduced odds of having MASLD [198] (however, it included only one study with a mean participant age over 65 years), which has also been shown in a more recent study in older persons [199]. Importantly, its improvements in MASLD are at least partly independent of overall weight loss [200]. These results have been confirmed in prospective trials specifically in MASLD, demonstrating benefits to the Mediterranean Diet alone [201] as well as a polyphenol enriched Mediterranean Diet [202], and higher adherence to the Mediterranean Diet has also been shown to improve the HSI (a surrogate for hepatic steatosis) in the PREDIMED study [203]. Other dietary options have been explored, including time restricted eating [192,204] and a low-carbohydrate, high-fat diet [205], although there is significantly less research on these strategies compared with the Mediterranean Diet. Conversely, there is a relationship between higher intakes of ultra-processed foods (UPFs) and an increased prevalence of MASLD in older persons [199]. It remains uncertain if this is related to co-morbid frailty [206] rather than a true causal link. Importantly, while UPFs are positively associated with MASLD, it has not yet been shown that reducing their intake would improve MASLD incidence or outcomes. In the context of older persons, UPF reduction should be undertaken cautiously, to ensure appropriate non-UPF substitution.

This nuance is especially relevant in older persons. There are significant additional co-morbidities in older persons living with MASLD compared to middle-aged people. As such, a primary focus on weight loss through diet may carry greater risk in this cohort, and while focusing instead on improving dietary patterns (e.g., shifting towards a more Mediterranean Diet pattern) may be a reasonable alternative goal, both are difficult to implement in practice for several reasons. Firstly, older adults with MASLD are more likely to be pre-frail or frail [124] as well as to develop persistent physical disability [138], and significant weight loss is an independent risk for the development of frailty [207,208]. Still, it should be noted that there is evidence to suggest that some obese older adults may functionally benefit from weight loss, particularly when combined with exercise [209]. Secondly, the Mediterranean Diet—beyond its utility for MASLD—is associated with a reduced risk of frailty in older persons [206,210,211]. Additionally, it is associated with reduced all-cause mortality, CVD, and improved quality of life [212,213,214,215]. Thirdly, food insecurity is common in older adults [216] and is associated with reduced adherence to a Mediterranean Dietary pattern of eating [217] as well as both MASLD and advanced hepatic fibrosis in the middle-aged [218]. Finally, multiple potential barriers are often noted in the implementation of dietary and lifestyle change, including psychosocial barriers [219], functional/motivational/financial issues [220,221,222], and diet palatability [223]. Understanding the optimal dietary therapy (ies) for older persons with MASLD requires further study.
**Recommendations for Future Research**Longitudinally assess the impact of (a) weight loss and (b) Mediterranean Diet adherence on older persons with MASLD in terms of both hepatic steatosis as well as functional outcomesLongitudinally assess the impact of the above on liver-related and extra-hepatic outcomes in older adults with MASLDAssess the barriers and facilitators to dietary modification in older persons with MASLD


### 6.3. Lifestyle Therapies: Physical Activity and Exercise

Similar to the recommendations for weight loss and dietary modification, both North American and European guidelines focus on exercise as a key tenet in the management of MASLD [17,91]. Sedentary behaviour is a known risk factor for MASLD [101,224,225], and conversely, higher levels of non-exercise physical activity (daily activities that involve movement without planning or strict control of exertion) is associated with both a lower risk for MASLD [225] as well as for both all-cause mortality and CVD in the general population [226,227]. Physical activity in MASLD specifically is also known to reduce all-cause mortality [228]. Recommended exercise targets for MASLD are >150 min/week of moderate intensity or >75 min/week of vigorous intensity exercise [17]. Once more, however, this is based on heterogeneous data, with a paucity of studies in older persons [229,230,231,232]. This lack of relevant data is akin to the lack of evidence underpinning the overall healthy physical activity guidelines for adults aged over 65 years in both the USA and Australia [233,234].

Importantly, physical activity is associated with a reduced risk of frailty [235], symptoms of frailty [236], and nursing home placement in older adults [237]. Whilst there are limited data on the impact of a physical activity intervention in older persons with MASLD, it is likely that—given the relationship between MASLD, frailty, sarcopenia, and myosteatosis [124,238]—a targeted intervention may improve not only hepatic steatosis but also the known associated physical dysfunction. This is particularly important in the setting of weight loss. If reducing bodyweight is considered an appropriate therapeutic goal for an overweight older person, the institution of calorie/kilojoule restriction in combination with resistance training has been shown to markedly attenuate weight loss-induced lean mass loss [239]. Older-person specific interventions that recognise and attempt to ameliorate the known barriers to exercise with ageing [114,222] may improve both liver and non-liver related outcomes as well as improve health span.
**Recommendations for Future Research**Assess the type, level and duration of exercise required to ameliorate poorer outcomes in older persons with MASLD, with a particular focus on frailty, persistent physical disability, and quality of life


### 6.4. Directed Pharmacotherapies

#### 6.4.1. Incretin Mimetics

In addition to the lifestyle recommendations of diet and exercise, adjunctive weight loss pharmacotherapy is becoming increasingly common in the treatment of overweight/obesity [240]—with an associated decrease in the need for bariatric surgery [241]. Despite the lack of a specific glucagon-like peptide (GLP)-1 target in hepatocytes, GLP-1 agonists may have additional liver-specific benefits via improvement in insulin resistance, managing co-morbid T2DM, and improving lipotoxicity [242,243]. These actions are particularly important in MASLD, both because of the known link between T2DM and an increased risk of progressive liver disease [14,244], as well as the known challenges of sustaining weight loss through diet and exercise alone [193,245]. A recent study compared lifestyle alone against GLP-1 agonist therapy in MASLD and showed comparable reductions in liver fat content, although improvements in glucose homeostasis were only seen in the GLP-1 agonist group [246].

Given this, it is no surprise that promising data are emerging to showcase the benefits from incretin monotherapy or combination therapy (a GLP-1 agonist combined with another incretin or glucagon) prescription in middle-aged adults with MASLD on a range of liver-related parameters including histology, transaminases, and MRI-quantified liver fat content [247,248,249,250,251]. Recently, the final data from the ESSENCE randomised controlled trial of 2.4 mg semaglutide versus placebo in MASH showed marked improvements in both steatohepatitis and liver fibrosis compared with placebo [252]. Additionally, observational data suggest a reduction in liver-related outcomes and reduced mortality in individuals with cirrhosis and T2DM treated with GLP-1 agonists (although not necessarily only due to MASLD) [253,254]. While earlier studies focused solely on those with compensated liver disease, a recent small pilot trial of survodutide (a GLP-1/glucagon dual-agonist) was safe and efficacious in decompensated cirrhosis [255]. Additionally, recent Taiwanese data show that there is a reduced risk of progression to cirrhosis amongst patients with T2DM compared to those using long-acting insulin [256] amongst middle-aged adults. Further data on the current role of GLP-1 receptor agonists can be found in a review article in Gut [257].

There are less data on the safety and efficacy of these agents in older adults with MASLD, although multiple studies have investigated their use in older adults in other conditions. A recent systematic review showed that GLP-1 agonists led to a reduced risk of MACE in adults ≥65 years [258], which is of particular interest given the previously described link between MASLD and MACE [128,129]. However, as previously mentioned, weight loss in older adults may present a double-edged sword, as it may be associated with an increased risk of frailty [207], or in the obese, may be protective [209]. This is especially concerning in older persons with MASLD, given the previously identified links with frailty and physical disability [124,138]. The addition of exercise training to incretin therapies to mitigate lean mass loss is an as-yet poorly studied area in the general literature [259], but may be important in the safe utilisation of these therapies in older persons to minimise potential harms. To our knowledge, there is no specific study investigating physical function outcomes in older persons with MASLD treated with incretin mimetics, nor are there data to guide decision making around BMI strata that may most benefit from such therapies in older persons. Additionally, while therapy cessation is known to lead to weight regain, there are less data on this in older persons [260], especially if the distribution of that gain is disproportionately adipose tissue, or whether there is a return of lean mass.
**Recommendations for Future Research**Evaluate the safety of incretin mimetics in older adults with MASLD in terms of body composition as well as changes associated with weight regain with medication cessationEvaluate whether incretin mimetics improve (a) liver-related and (b) non-liver related cardiometabolic outcomes in older adults with MASLDEvaluate whether incretin mimetics improve physical function in older adults within various BMI strataDescribe changes in dietary intake in older adults prescribed incretin mimetics


#### 6.4.2. Thyroid Hormone Receptor-β Agonists

In 2024, the Federal Drug Administration (FDA) in the USA approved resmetirom [189] as the first MASH-specific agent licensed for use, due to its Phase III data showing improvements in both steatohepatitis and fibrosis in at-risk disease [261]. Previous data had established a relationship between altered thyroid hormone metabolism and MASLD-related inflammation and fibrosis [262]. Resmetirom is an orally active liver-specific compound that preferentially binds to thyroid hormone receptor beta (THR-β), the predominant receptor isoform in hepatocytes. This reduces T4 without impacting thyroid stimulating hormone (TSH) or the more biologically active triiodothyronine (T3) [262]. Taken together, these actions improve lipid metabolism, mitochondrial β-oxidation, reduce VLDL and triglycerides, and reduce liver stiffness and fat content, all with relatively minor deleterious impacts on other tissues, such as the heart and bone density [261,262,263]. In addition to these key cardiometabolic benefits, secondary outcomes suggest that resmetirom improves health-related quality of life in MASLD [264]. Even more potent THR-β agents are in development and may have improved efficacy over resmetirom, although these trials are ongoing [262]. It is uncertain whether resmetirom or other THR-β agents will have any role in MASLD-related cirrhosis.

Again, there are no data on the use of resmetirom in older persons. This is relevant given the links between ageing and thyroid dysfunction generally and the data on the impact of thyroid status on cognitive function [265], as well as the increased resistance to thyroid hormones in older persons [266]. Resmetirom can cause gastrointestinal side-effects (including diarrhoea and nausea) [261], and currently the impact of this on dietary patterns and overall nutritional intake is unknown. Pharmacologically, resmetirom is hepatically metabolised as a CYP2C8 substrate, which is the same cytochrome P450 enzyme that metabolises a number of commonly prescribed medications, including certain non-steroidal anti-inflammatory drugs, some statins, and amiodarone, as well as being inhibited by clopidogrel and gemfibrozil [267,268,269]. Older persons are more likely to be prescribed multiple agents [270], and the impact of polypharmacy on the efficacy of resmetirom and other agents remains understudied.
**Recommendations for Future Research**Evaluate whether resmetirom has a differential impact on thyroid function parameters in older persons with MASLDEvaluate whether the gastrointestinal side-effects of resmetirom lead to an undesirable dietary shift in older persons with MASLDEvaluate the potential polypharmacy and co-prescribing risks to older persons with MASLD being offered Resmetirom


#### 6.4.3. Fibroblast Growth Factor-21

While resmetirom was the first FDA licensed therapy for MASH, other anti-fibrotic agents are currently undergoing Phase II trials. Fibroblast Growth Factor 21 (FGF21) is associated with obesity, dyslipidaemia, pericardial fat, and insulin resistance, with a half-life of approximately 0.5–2 h [271,272]. Pegozafermin is a long-acting recombinant FGF21 analogue with a half-life of 55–100 h that has been shown to improve insulin sensitivity and decrease de novo lipogenesis, hypertriglyceridaemia [273], hepatic fibrosis and MASH [274]. One trial showed up to 27% improvement in fibrosis stage over 24 weeks (compared to 7% for placebo) in non-cirrhotic middle-aged adults. While these data need to be confirmed to be sustained over a longer duration and evaluated in larger studies, pegozafermin is a promising future option for at-risk MASH.

Efruxifermin, another FGF21 analogue, has been studied in MASH with fibrosis stage 2 or 3 (but not cirrhosis) with similarly promising results, including improvements in fibrosis stage (without worsening of MASH) as well as improvements in non-invasive markers of liver injury (such as transaminase levels and ELF score) in the HARMONY study [275]. Preliminary data presented in abstract form of up to 96 weeks of use confirmed ongoing improvements in both steatohepatitis and fibrosis, including in those participants who did not respond in the initial 24-week period [276]. Another Phase II study has demonstrated safety with signals of efficacy in patients with compensated cirrhosis, expanding the potential population that may benefit from efruxifermin [277]. While early published data from the SYMMETRY study showed no improvement in fibrosis stage at 36 weeks, 50 mg efruxifermin appeared to reduce fibrosis in subjects with cirrhosis at 96 weeks [278]—further work evaluating longer-term use of FGF21 analogues is expected.

Like resmetirom, there are GI related side-effects from FGF-21 analogues, though, once more, these are inadequately studied in older adults [274,275,277]. Importantly for older patients, there have also been safety concerns raised regarding bone health, with accelerated bone loss in some (but not all) of the studies to date [188,278].
**Recommendations for Future Research**Evaluate the efficacy of FGF21 agonists in older persons with MASLDDetermine whether accelerated bone loss in those with and without pre-existing osteopenia or osteoporosis occurs with FGF21 agonist use in older personsEvaluate the impact of gastrointestinal side-effects on the dietary patterns of older persons with MASLD


#### 6.4.4. Co-Prescriptions and Combinations

Given the potential synergies between different therapeutic options, early work has begun to investigate the potential combinations of therapies with differing mechanisms of action. One recent study showed an additive benefit of efruxifermin to GLP-1 agonism in terms of hepatic fat fraction reduction and lipid parameters compared to GLP-1 monotherapy in participants with T2DM [279], and combination therapies have been recently editorialised as the “next frontier in MASH” [280]. While relatively few participants in the landmark resmetirom MAESTRO-NASH study were taking GLP-1 agonists at baseline (and thus co-prescription is difficult to endorse), the AASLD has recently noted that individual clinical judgement can be applied when considering the use of both therapies concurrently [269].
**Recommendations for Future Research**Evaluate whether co-therapy is synergistically beneficial for fibrosis reduction as well as metabolic risk management in (a) middle-aged persons and (b) older adults specifically


### 6.5. Pharmacotherapies: Co-Morbidity Management

It is well known that participants with MASLD are more likely to have, and to develop, T2DM [96,281,282], dyslipidaemia [105], and renal impairment [283,284,285]. Further, these diseases are more common in older adults than in the general population, despite the need to re-evaluate ‘normal’ cut-offs with aging [286,287,288,289]. Some of the therapies used to manage these conditions may have indirect benefits in MASLD, and while they are not licensed for liver-specific indications, their continued prescription may have ancillary benefits.

Historically, there has been caution around the use of statins in individuals with MASLD or other chronic liver diseases, due to their perceived risk of precipitating acute liver injury [290]. However, this has been largely debunked, with numerous contemporary studies supporting the initiation of statins in those with various liver diseases or elevated transaminase levels [291,292]. In addition to their role in the management of dyslipidaemia and the primary and secondary prevention of CVD, statins may reduce the risk of both MASLD development, as well as protect against fibrosis in those with existing MASLD [293]. Additionally, they may reduce portal pressure in cirrhosis and reduce the risk of hepatocellular carcinoma development [294,295]. Nevertheless, statin use in older persons can be complicated by concerns around increasing the risk of T2DM development [296] and potential inefficacy or reduced efficacy when used for primary prevention of CVD [297,298]. While statin therapy may have benefits in persons with MASLD, their relative risks and benefits in older persons warrants further study; STAREE (Statins in Reducing Events in the Elderly) trial results are keenly awaited [299].

Similarly, from a cardiovascular and preventive health perspective, aspirin for use as primary prevention remains a contentious issue. The landmark ASPREE study showed no net benefit of aspirin in older adults for MACE, disability-free survival, or all-cause mortality [300,301,302] (although it did show benefit in reducing the incidence of T2DM [303]), and a post-hoc analysis showed no benefit on mortality or MACE in older persons with MASLD [128]. However, existing data do show potential benefits of aspirin in terms of reducing MASH and rates of fibrosis progression [304,305], as well as a marked reduction of hepatocellular carcinoma rates in mixed-aetiology chronic liver disease [306] and MASLD [307]. Given the risks of excess bleeding with aspirin use in older adults [301,308], careful patient selection is important if considering aspirin for chemoprevention or treatment for MASLD and MASLD-specific complications in older persons.

In terms of the management of co-morbid T2DM, sodium-glucose co-transporter 2 inhibitors (SGLT2i) are commonly used to not only improve glycaemic control but to reduce the risk of incident kidney disease and manage co-morbid heart failure [309,310]. These agents may also confer benefits for MASLD [311], including on histologic endpoints [312]. When considering their role in preventing outcomes for patients with coexistent T2DM and MASLD, recent data are supportive—a Japanese study showed that SGLT2i were superior to DPP4 inhibitors for the reducing the development of varices and malignancy [313], and a Korean study has shown that SGLT2i therapy is associated with both higher rates of MASLD regression and lower rates of adverse liver outcomes than other oral antidiabetic therapies [314]. Importantly, SGLT2i are considered safe in older adults [315,316], and thus the existing data support consideration of therapy with SGLT2i for older adults with co-morbid MASLD and T2DM, especially if GLP-1 agonists are not tolerated or available. A key caveat is the diuretic and blood pressure lowering effects of SGLT2i, with a potential for increased risk of falls [317,318]. Frailty assessment (given its relative frequency in MASLD [124]) may assist in identifying subgroups of older persons most likely to benefit with lower risk.
**Recommendations for Future Research**Determine whether any subgroup(s) of older participants, particularly in relation to frailty and disability status, may particularly benefit from the potential MASLD reduction associated with aspirin and/or statin therapyDetermine whether older persons with combination T2DM and MASLD have an outsized benefit from SGLT2 inhibition compared to other therapies, and whether stratifying by frailty identifies a subgroup more likely to benefit


## 7. Economic Impacts

MASLD, as a major clinical concern and public health burden, represents a significant cost to the health system. A study evaluating the economic impact of MASLD in the US and parts of Europe (Germany, France, Italy, and the United Kingdom) found projected annual costs to be as high as USD 292.19 billion and EUR 227.84, respectively [319]. A more recent study in Italy highlighted excess potential direct costs of EUR 1.77 billion compared to those without MASLD, and up to EUR 7.94 billion including modelled flow-on effects (including increased rates of cancer and T2DM but excluding quality of life impacts) [320]. Importantly, it is likely that those economic costs are most significant in the older adult population; in the US, the per-patient costs were calculated to be highest in the ≥65 year old group [319].

In the context of the developing literature highlighted in this review, it is also important to note that many of these models do not necessarily directly account for the impacts of frailty or physical disability. In the general ageing literature, it is well recognised that frailty is costly [321,322,323]. However, the interplay between MASLD and ageing and the consequent potential increase in older persons with frailty, and at risk of persistent physical disability, is understudied from a health economic perspective. Consequently, the costs identified in MASLD modelling studies could be significantly underestimated in relation to older adults.

Similarly, when considering pharmacoeconomic modelling for the benefits of potential MASLD therapies, studies often do not consider the impact on older persons specifically [324], and we do not yet know the impact of modern pharmacotherapeutic options on frailty or the progression to disability in older persons with MASLD. As highlighted in Section 6, future work to better understand the risks and benefits of the novel therapies for MASLD in older persons will also help guide more accurate health economic modelling.
**Recommendations for Future Research**Determine the health economic burden of MASLD in older adults including both direct and indirect costsDetermine whether modern pharmacotherapies are cost-effective for treating MASLD in older persons


## 8. Conclusions

MASLD is a commonly identified condition in older persons and, like in the middle-aged, is strongly associated with multiple other cardiometabolic conditions. However, the treatment paradigm may differ significantly. A greater focus on frailty and functional assessment, appropriate selection of adjunctive therapies for co-morbid conditions, and consideration of polypharmacy and the competing risks inherent to ageing are important and understudied areas; focusing solely on weight loss may not yield the benefits we and our patients seek. When counselling older persons with MASLD in the clinic, highlighting these links, and gaps in the literature, will be important, and shifting the focus away from mortality towards improving health span will likely realign expectations for both clinicians and patients. Further research into understanding the impact of MASLD on older persons and how best to care for them is crucial, as the epidemiology and risks differ compared to younger and middle-aged persons, while, at the same time, the population continues to age rapidly.

## Figures and Tables

**Table 1 nutrients-17-02189-t001:** Components of selected steatosis indices. Adapted from Clayton-Chubb et al. [63].

Score Components	FLI	FSI	DSI	HSI	ZJU	VAI	LAP
BMI (kg/m^2^)	**✓**	**✓**	**✓**	**✓**	**✓**	**✓**	
Abdominal circumference (cm)	**✓**					**✓**	**✓**
GGT (U/L)	**✓**						
Triglycerides (Yes/No)	**✓**	**✓**	**✓**		**✓**	**✓**	**✓**
ALT (U/L)		**✓**	**✓**	**✓**	**✓**		
AST (U/L)		**✓**		**✓**	**✓**		
Age (years)		**✓**	**✓**				
Sex (Male/Female)		**✓**	**✓**	**✓**	**✓**	**✓**	**✓**
Hypertension (Yes/No)		**✓**	**✓**				
Diabetes (Yes/No)		**✓**	**✓**	**✓**			
Serum glucose			**✓**		**✓**		
Ethnicity			**✓**				
HDL cholesterol						**✓**	

BMI: Body Mass Index; GGT: gamma glutamyltransferase; ALT: alanine aminotransferase; AST: aspartate aminotransferase; FLI: Fatty Liver Index; FSI: Framingham Steatosis Index; DSI: Dallas Steatosis Index; HSI: Hepatic Steatosis Index; ZJU: ZJU Index; VAI: Visceral Adiposity Index; LAP: Lipid Accumulation Product.

**Table 2 nutrients-17-02189-t002:** Components of selected liver fibrosis scores.

Score Components	FIB-4	BARD	APRI	NAFLD Fibrosis Score	BIMAST	SAFE	MAF-5	ELF
Age	**✓**			**✓**		**✓**		
ALT (U/L)	**✓**	**✓**		**✓**		**✓**		
AST (U/L)	**✓**	**✓**	**✓**	**✓**	**✓**	**✓**	**✓**	
Platelets	**✓**		**✓**	**✓**		**✓**	**✓**	
BMI		**✓**		**✓**	**✓**	**✓**	**✓**	
Waist Circumference							**✓**	
Diabetes (Yes/No)		**✓**		**✓**		**✓**	**✓**	
Impaired Fasting Glucose (Yes/No)				**✓**				
Albumin				**✓**				
Globulins						**✓**		
Tissue inhibitor of metalloproteinases 1 (TIMP-1)								**✓**
Amino-terminal propeptide of type III procollagen (PIIINP)								**✓**
Hyaluronic acid (HA)								**✓**

BMI: Body Mass Index; ALT: alanine aminotransferase; AST: aspartate aminotransferase; FIB-4: Fibrosis-4; BARD: BMI, AST/ALT Ratio, Diabetes; APRI: AST to Platelet Ratio Index; BIMAST: BMI and AST; SAFE: Steatosis-Associated Fibrosis Estimator; MAF-5: Metabolic Dysfunction-Associated Fibrosis-5; ELF: Enhanced Liver Fibrosis.

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
