# Peer review of "Steatotic Liver Disease in Older Adults: Clinical Implications and Unmet Needs"

_nutrients, 2025, doi:10.3390/nu17132189_

Round 1

Reviewer 1 Report

Comments and Suggestions for Authors

Dear authors

Congratulations on an excellent article. Very detailed, useful for clinical practice, with numerous references.Fully corresponds to the topic of a special issue.

In order to improve quality of the article I recommend some suggestions:

Please kindly state in the materials and methods section clear inclusion criteria for the inclusion of the mentioned references, as well as the period from when to when they were collected.
Also state MESH terms or keywords and clear exclusion criteria.

Please, in a separate section, write about the economic repercussions of this pathology in the elderly.
I think it is a very high-quality review.
It fully corresponds to the theme of the special issue.
Since the paper has no images or tables, I suggest that the authors create a graphic summary.
This paper, in my opinion, requires minor corrections.
The authors are native speakers and no English corrections are necessary.

Author Response

Please see attached response to reviewers -- we've included responses to all 3 to make understanding the responses and manuscript decisions easier. Thank you. 

Reviewer 2 Report

Comments and Suggestions for Authors

The manuscript entitled Steatotic Liver Disease in Older Adults: Clinical Implications and Unmet Needs is a narative review. It describes the prevalence, clinical and sociodemographic associations, and outcomes for older persons with Steatotic Liver Disease. It is an interesting narative review because this pathology is relative frequently.

I have some comments/suggestions for the authors.

Why the authors used the  Recommendations for Future Research after some chapters ? Why are these recommendations only in certain chapters (e.g. 4.1)?Why do the authors believe that these recommendations should be taken into account? It would be much more useful to make a table with the data from the literature and the results obtained so far on the respective topic.

In chapter 4 entitled MASLD and clinical outcomes in older persons, I suggest that 4.1. MASLD and Mortality to be placed after other comorbidities associated with MASLD. Comorbidities associated with MASLD might be listed in Chapter 4 in order of frequency of association.

The number of references (n=295) is too many (references are a half of manuscript pages). Please make a rigourous selection of references.

Author Response

(The authors gave the same response as above.)

Reviewer 3 Report

Comments and Suggestions for Authors

This is a well-written review paper on ageing and MASLD. To improve the comprehensiveness of the paper, I suggest adding two additional sections.

The first could focus on diverticulosis and MASLD. Since diverticulosis is one of the most common gastrointestinal diseases in the ageing population, and given the shared risk factors, this relationship should be further elaborated upon (see: https://pubmed.ncbi.nlm.nih.gov/33951119/).

The second section could address infectious complications. The relationship between NAFLD and infections has been previously explored, and the authors should review the relevant literature, particularly regarding how immunosenescence and MASLD interact.

Optionally, a third paragraph discussing age-related changes in the gut microbiota and their influence on MASLD development in the elderly would also be a valuable addition.

Comments on the Quality of English Language

no issues

Author Response

(The authors gave the same response as above.)

Round 2

Reviewer 3 Report

Comments and Suggestions for Authors

I would like to thank the authors on detailed revisions. I would recommend the acceptance in its current form.